# Comparison of Patients Monoinfected with Hepatitis C Virus and Coinfected with Hepatitis B/C in the Amazon Region of Brazil

**DOI:** 10.3390/v14050856

**Published:** 2022-04-21

**Authors:** Regiane M. A. Sampaio, Paola Eduarda F. Dantas, Maria Inês C. da Silva, Joseane R. da Silva, Patrícia F. Nunes, Amanda C. Gomes, Luisa C. Martins

**Affiliations:** 1Laboratory of Clinical Pathology of Tropical Diseases, Federal University of Pará (UFPA), Tropical Medicine Center (NMT), Umarizal, Belém-Pará 66055-240, Brazil; regianearnund@yahoo.com.br (R.M.A.S.); paolaf.dantas@gmail.com (P.E.F.D.); mariaines_caricchio@hotmail.com (M.I.C.d.S.); joseanefarmaceutica@hotmail.com (J.R.d.S.); dsnpatriciaferreira@gmail.com (P.F.N.); 2Institute of Health Sciences, School of Pharmacy, Federal University of Pará (UFPA), Belém-Pará 66075-110, Brazil; 3Graduation in Medicine, University Center of the State of Pará (CESUPA), Belém-Pará 66613-903, Brazil; amandacaricio@uol.com.br

**Keywords:** Amazon region, Brazil, coinfection, HBV, HCV

## Abstract

Hepatitis B and C are the most common causes of liver disease worldwide. The two infections share many similarities such as a global distribution, the same routes of transmission, hepatotropism, and the ability to cause chronic infection. The consequences of HBV/HCV coinfection are still being studied. The aim of this study is to describe and compare the epidemiological and laboratory profile and the degree of hepatic fibrosis between HCV-monoinfected and HBV/HCV-coinfected patients in the Brazilian Amazon region. ELISA tests were used for the investigation of HBV and HCV serological markers, and molecular tests were used for the detection and genotyping of these viruses. Additionally, transaminases were measured, and a FibroScan was performed for the analysis of liver function. A total of 328 patients with HCV participated in the study. The serological prevalence of HCV/HBV coinfection was 10.77%. A comparison of risk factors between the monoinfected and coinfected groups showed that illicit drug use, sharing sharp instruments, and tattooing/piercing are significantly associated with coinfection. The monoinfected patients had a higher HCV load than the coinfected patients. A viral interaction was observed in this study in which the presence of a coinfection with HBV appears to influence HCV replication. Further studies are necessary to better understand this interaction.

## 1. Introduction

Hepatitis B and hepatitis C are the most common causes of liver disease worldwide. These infections pose major public health challenges since they affect about 325 million people and cause 1.34 million deaths every year. An estimated 292 million people worldwide are living with chronic hepatitis B, and 71.1 million have chronic hepatitis C [1,2,3].

According to the Brazilian Ministry of Health, 212,031 cases of hepatitis B and 182,389 cases of hepatitis C have been registered with the National Notifiable Diseases Information System (SINAN in the Portuguese acronym) between 1999 and 2016 in the northern region of the country, corresponding to a prevalence of 14.2% and 3.1%, respectively [4]. In parallel, studies indicate the existence of areas with a silent or occult prevalence of these infections in the northern region of the country, which is either due to failures in the notification system or to the small number of published studies and difficult access to some areas [5].

Hepatitis C is caused by the hepatitis C virus (HCV), which belongs to the family Flaviviridae and whose genetic material consists of a positive-sense single-strand RNA molecule of 9600 kb [6]. Hepatitis B is caused by the hepatitis B virus (HBV), which belongs to the family Hepadnaviridae and whose genetic material consists of a a double-chained circular DNA, measuring 3.2 Kb, which replicates through a process that involves an RNA intermediate and reverse transcription [7].

Infections with HBV and HCV share many similarities such as a global distribution, the same routes of transmission, hepatotropism, and the ability to cause chronic infection that can lead to cirrhosis and hepatocellular carcinoma [8]. Within this context, studies indicate that dual infection with HBV/HCV is not uncommon and might be associated with an increased risk of hepatocellular carcinoma and a poorer clinical prognosis when compared to monoinfection with HBV or HCV [8,9,10]. Coinfection with these viruses is relatively common in endemic areas and among individuals at high risk of parenteral transmission. Studies have reported a prevalence of coinfection ranging from 2 to 10% among anti-HCV-positive individuals and from 3 to 20% among HBsAg-positive patients [10,11,12].

In HBV/HCV co-infection, viral serologies vary depending on whether the co-infection is simultaneous or a superinfection, but HCV infection is almost always evidenced by detection in the serum of HCV RNA and antibodies against the infection (anti-HCV) [13], while HBV infection can be evident or occult. In this scenario, the prevalence of coinfection with occult hepatitis B may be underestimated, with the patients being negative for HBsAg and having low viral detection in blood and/or tissue. In the case of occult HBV infection, HBV DNA is present in HBs-Ag-negative individuals with or without the serological markers of prior infection, i.e., antibodies against HBsAg (anti-HBs) and nuclear proteins (anti-HBc) [14].

The dominance of one virus over the other and their subsequent evolution are influenced by viral and host factors. Important determinants include the sequence of viral infections, the replication capacity of the strains, the presence of mutations in the viral genome, the virus-induced immune response, and the hepatocellular reaction [15].

In Belém, the capital of the state of Pará, coinfection with HBV and HCV has been little studied. The aim of the present study is to determine the prevalence of HBV/HCV coinfection and to compare epidemiological and laboratory data and the degree of hepatic fibrosis between patients with HCV monoinfection and HBV/HCV coinfection.

## 2. Materials and Methods

### 2.1. Study Population

The participants in this study were registered with the Viral Hepatitis Service of the Tropical Medicine Center, Federal University of Pará, a referral center for viral hepatitis infections in the city of Belém, State of Pará, Brazil. We selected 328 patients with HCV who had a reactive anti-HCV ELISA result, were positive for HCV RNA, had not undergone treatment, and were negative for HIV (non-reactive anti-HIV ELISA). This cross-sectional, analytical, descriptive study was conducted from December 2016 to May 2019.

Approximately 10 mL of peripheral blood was collected from each patient for the investigation of HBV infection, determination of HCV load, and measurement of transaminases. The study was approved by the Ethics Committee on Research Involving Humans of the Tropical Medicine Center, Federal University of Pará (approval number 2.432.635).

### 2.2. Epidemiological Data

The following epidemiological data were obtained from charts filled out during collection of the biological material: socioeconomic and sociodemographic data and risk factor such as illicit drug use, blood transfusion, number of sexual partners, and sharing of sharp instruments.

A standardized questionnaire regarding lifestyle habits (excessive alcohol consumption and smoking habits) was used. Average drinking levels were evaluated to calculate the amount of alcohol consumed: sugarcane rum = 40%; beer = 5%; wine = 12%; and other distilled beverages = 55%. The threshold of alcohol consumption was standardized by calculating the average daily intake in grams, with an acceptable dose of 70 g of alcohol per week for women and 140 g for men [16].

Smoking intensity was defined as the total number of cigarettes consumed per day, with each hand-rolled cigarette being equivalent to five manufactured cigarettes [17]. Subjects smoking more than five cigarettes per day were classified as smokers, and those smoking fewer than five cigarettes per day or who did not smoke at all were classified as non-smokers.

### 2.3. Assessment of Liver Elasticity (FibroScan Elastography)

Elastography was performed as described above [18]. All measurements were obtained in the right lobe of the liver through the costal space. The acquisition of images was guided by ultrasound and the measurement of the area was performed with a depth ranging from 25 to 45 mm; 10 valid measurements were obtained per patient. Results were expressed in kilopascals (kPa).

Liver stiffness corresponds to the median value of all valid measurements. The Metavir fibrosis stages were classified based on the kPa values as follows: F0 when 2.0 to 4.5 kPa, F1 when 4.5 to 5.7 kPa, F3 when 5.7 to 12.0 kPa, and F4 when 12.1 to 21.0 kPa.

### 2.4. Laboratory Methods

Serological tests: Third-generation enzyme-linked immunosorbent assays (ELISA) were used for the detection of HBsAg, anti-HBc, and anti-HCV, using commercial tests (Dia.Pro, Italy) according to manufacturer instructions.

Extraction and detection of HCV genetic material: In samples with anti-HCV-positive serology (ELISA), viral RNA was extracted from 200 µL of the patient’s serum with the QIAmp Viral RNA kit (Qiagen, Germany), following manufacturer instructions.

RNA investigation was performed by nested PCR, using primers that target the 5′-UTR region. The first reaction consisted of synthesis and amplification of cDNA in a single step using SuperScript™ III One-Step RT-PCR System with Platinum™ Taq DNA Polymerase kit, where a mixture was achieved containing 5 µL of 2X Reaction Mix, 1 µL of the k10 (5′-GGC GAC ACT CCA CCA TRR-3′) and k11 (5′-GGT GCA CGG TCT ACG AGA CC-3′) primers, 5 µL of DNAse-free ultrapure water, 5 µL of RNAse, and 1 µL of One-Step Taq DNA Polymerase (Invitrogen, São Paulo, Brazil) for a final volume of 25 µL. The samples were incubated in a thermocycler at 42 °C for 45 min. The amplification conditions were: initial denaturation at 94 °C for 2 min, followed by 35 cycles of denaturation at 94 °C for 30 s, annealing at 54 °C for 30 s, and extension at 72 °C for 45 s. A final extension was performed at 72 °C for 7 min, and the samples were cooled to 4 °C. The second reaction mixture contained 2.5 µL of 10X buffer, 4 µL of dNTPs, 1.5 µL of MgCl_2_, 1 µL of k15 (5′-ACC ATR RAT CAC TCC CCT GT-3′) and k16 (5′-CAA GCA CCC TAT CAG GCA GT-3′) primers, 12.5 µL of DNAse- and RNAse-free ultrapure water, and 0.5 of µL Platinum Taq DNA Polymerase (Invitrogen), added to the sample for its final volume of 25 µL.

The virus was genotyped by the restriction fragment length polymorphism (RFLP) technique using AvaII and RsaI restriction enzymes. Positive and negative controls were included in all reactions [19].

Quantification of HCV load: The Abbott RealTime HCV assay was used for the quantification of HCV RNA. The detection limit of the assay is 12 IU/mL of viral copies per mL serum.

Extraction and detection of HBV genetic material: In all samples with HBsAg- and/or anti-HBc-positive serology (ELISA), the HBV DNA was extracted from 200 µL of serum with the QIAamp DNA Mini kit (Qiagen, Germany). The HBV DNA was detected by identification of the S gene fragment [20]. All qPCR assays were run using a TaqMan Universal PCR Master Mix kit (Applied Biosystems) and contained control samples for the HBV DNA.

Measurement of liver enzymes: Aminotransferases (ALT and AST), gamma-glutamyl transferase, and alkaline phosphatase were measured for the evaluation of liver function. The enzymes were quantified in a semi-automatic TP Analyzer using commercial kits from In Vitro Indústria e Comércio Ltd.a, Minas Gerais, Brazil.

### 2.5. Statistical Analysis

Statistical analysis was performed with the Bioestat 5.0 program [21]. Odds ratios were calculated to compare risk factors between the two groups studied. The Mann–Whitney test was used for an inferential analysis of the laboratory results. A level of significance of 0.05 was adopted.

## 3. Results

A total of 328 patients with HCV who had reactive anti-HCV serology and were positive for HCV nucleic acid participated in the study. Serological markers for HBV were observed in 10.97% (36/328) of the participants; only HBsAg was detected in 36.11% of these patients (13/36), both HBsAg and total anti-HBc in 41.67% (15/36), and only anti-HBc in 22.22% (8/36).

HBV DNA was isolated from the samples of 88.89% (32/36) of the patients with serological markers for HBV. The four patients with positive serology (only anti-HBc) who tested negative for HBV DNA were excluded from the study. Thus, the monoinfected group included 292/324 (90.12%) patients who only exhibited markers for HCV, and the coinfected group consisted of the 32/324 (9.88%) patients in whom genetic material of both viruses (HCV RNA and HBV DNA positive) was detected. Regarding the viral genotypes, the HCV genotypes 1, 2, and 3 and the HBV genotypes A, D, and F were observed (Table 1).

The mean age of the 324 selected patients was 49.67 years (range 18 to 85 years). Regarding sex, 58.02% (188/324) of the participants were males, and 41.97% (136/324) were females. The comparison of the data points regarding sex and age between the monoinfected (HCV) and coinfected (HBV/HCV) patients showed no statistically significant difference.

A comparison of the epidemiological data between the monoinfected (HCV) and coinfected (HBV/HCV) patients is shown in Table 2. Among the risk factors associated with HBV/HCV coinfection, a statistically significant difference was observed for illicit drug use (*p* = 0.01), blood transfusion (*p* = 0.0103), and the presence of tattoos/piercings (*p* = 0.03).

With respect to the laboratory tests, comparison of the elastography data showed no difference in the degree of fibrosis between patients with HBV/HCV coinfection and monoinfection. Additionally, the mean transaminase levels did not differ significantly between the two groups. Coinfected patients had a lower HCV load than monoinfected patients (Table 3).

## 4. Discussion

Few Brazilian studies have analyzed coinfection with HBV and HCV, particularly in the northern region. In the present study, the prevalence of HCV/HBV coinfection was 9.88% (32/324) among patients with HCV. Among the coinfected patients, 12.5% (4/32) were negative for HBsAg with the presence of HBV DNA, suggesting the presence of occult HBV infection. Some studies suggest a higher prevalence of occult hepatitis B due to suppression of this virus by HCV [22,23,24]. Sant’Anna et al., found a prevalence of occult hepatitis B of 14.36% among patients with suspected HBV infection treated in the city of Belém, Pará [18].

Alternatively, a frequency of 40.62% of individuals who tested positive only for HBsAg was observed, which, according to the literature, constitutes an incubation period or early acute phase, where lower anti-HBC titers are observed to undetectable serological levels [25,26].

In the present study, a comparison of the different risk factors evaluated between HCV-monoinfected and HCV/HBV-coinfected patients showed that illicit drug use (*p* = 0.01), the presence of tattoos and/or piercing, and blood transfusion were significantly associated with coinfection. According to the Brazilian Ministry of Health and Oliveira-Filho et al., HCV is almost exclusively transmitted through the parenteral route by exposure to blood containing viral particles, as well as by sharing needles among illicit drug users or instruments used for piercing and tattooing [2,27].

In the Amazon region, a high prevalence of HCV infection is observed among illicit drug users [27]. According to Tyson et al., drug users are at increased risk of HCV/HBV coinfection [10]. Additionally, some studies have demonstrated a correlation between a history of blood transfusion and an increased number of hospitalizations and HCV/HBV coinfection [10,11,28].

Although both HBV and HCV replicate in hepatocytes, these viruses have very different life cycles. HBV is a DNA virus that replicates in the nucleus of hepatocytes, while HCV is an RNA virus that replicates exclusively in the cytoplasm of liver cells. Studies have demonstrated that these viruses interact during coinfection and that this interaction may reduce viremia of one or both viruses and increase damage to the host [13].

The present study shows a lower HCV load in patients coinfected with HCV/HBV compared to HCV-monoinfected patients (619,000 and 1,815,000 IU/mL, respectively). In a study conducted in Europe, Marot et al. [29] also observed changes in the viral load, demonstrating a higher HCV load in monoinfected patients compared to dual-infected patients, which suggests that in co-infection there is suppression of HCV replication by the HBV superinfection, the which can result in a state of latent HCV infection.

Comparison of the degree of fibrosis in this study revealed no significant difference between the monoinfected and coinfected patients. The association between coinfection and increased liver damage is contradictory. Some studies have demonstrated an association of HCV/HBV coinfection with increased liver damage and the risk of developing cirrhosis and hepatocellular carcinoma [8,9,28], while others do not report this association [28,29]. Cardoso et al. did not find any association between occult HBV infection and clinical picture, virological, and histological data, and their work suggests that cohort studies should be performed to identify the real impact of occult HBV in patients with liver disease [30].

## 5. Conclusions

An interaction between these viruses was observed in the present study, in which the monoinfected patients had a larger number of HCV RNA copies than the coinfected patients. There was no association of coinfection with increased liver damage despite a slight increase in fibrosis and transaminase levels, which, however, were not statistically significant. Further studies are necessary to better understand this interaction, as well as the analysis and comparison with a group of HBV-monoinfected patients.

## Figures and Tables

**Table 1 viruses-14-00856-t001:** Laboratory investigation of HCV and HBV in the study participants.

HBV/HCV Diagnosis	HCV Monoinfected (n = 292)	%	HBV/HCV Coinfected (n = 32)	%
HBsAg	-		13	40.62
Anti-HBc	-		4	12.5
HBsAg + anti-HBc	-		15	46.88
HCV genotype				
1	164	56.16	17	53.12
1a	2	0.68	1	3.12
1b	42	14.39	6	18.76
2	30	10.28	4	12.5
3	54	18.49	4	12.5
HBV genotype				
A	-		27	84.37
D	-		2	6.25
F	-		3	9.37

**Table 2 viruses-14-00856-t002:** Comparison of epidemiological data between HCV-monoinfected and HBV/HCV-coinfected patients.

Variable	HCV Monoinfected (n = 292)	%	HBV/HCV Coinfected (n = 32)	%	OR (CI)	*p*
Tobacco use						
Yes	4	1.37	2	6.25	0.20	0.21
No	288	98.63	30	93.75	(0.03–1.18)	
Alcohol consumption						
Yes	10	3.40	2	6.25	0.51	0.75
No	282	96.60	30	93.75	(0.11–2.54)	
Illicit drug use						
Yes	10	4.43	6	18.75	6.50	0.01
No	282	96.57	26	81.25	(2.19–19.33)	
Age at first sexual relation						
≤12 years	16	5.47	2	6.25		
13 to 17 years	196	67.12	18	56.25		
≥18 years	80	27.39	12	37.50		
Number of sexual partners in one year						
1 to 2	270	92.46	28	87.50	0.57	0.52
3 to 5	22	7.54	4	12.50	(0.18–1.77)	
Condom use						
Never/sometimes	242	82.87	24	75.00	0.61	0.38
Always	50	17.13	8	25.00	(0.26–1.45)	
Occurrence of STI						
Yes	52	17.80	10	31.25	0.47	0.11
No	240	82.20	22	68.75	(0.21–1.06)	
Blood transfusion						
Yes	32	10.95	12	37.50	4.87	0.01
No	260	89.05	20	62.50	(2.18–10.89)	
Tattooing/piercing						
Yes	30	10.27	8	25.00	2.91	0.03
No	262	89.73	24	75.00	(1.20–7.05)	
Sharing sharp instruments						
Yes	64	21.61	10	31.25	1.61	0.33
No	228	78.38	22	68.75	(0.72–3.59)	

Abbreviations: STI, sexually transmitted infections.

**Table 3 viruses-14-00856-t003:** Comparison of viral load, elastography, and transaminase levels between HCV-monoinfected and HBV/HCV-coinfected patients.

Laboratory Test	HCV Monoinfected (n = 292)	HBV/HCV Coinfected (n = 32)	*p*
Liver stiffness (kPa)	6.53 ± 3.2	7.28 ± 2.7	0.31
AST (U/L)	69.08 ± 1.79	85.54 ± 1.82	0.26
ALT (U/L)	75.47 ± 1.83	86.93 ± 1.96	0.38
HCV RNA (log IU/mL)	1815.88 ± 15.82	619,875 ± 20.17	0.01

Values are the mean ± standard deviation. Mann–Whitney *U* test. Abbreviations: ALT, alanine aminotransferase; AST, aspartate aminotransferase.

## Data Availability

Not applicable.

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
