# Peer review of "Comparison of Patients Monoinfected with Hepatitis C Virus and Coinfected with Hepatitis B/C in the Amazon Region of Brazil"

_viruses, 2022, doi:10.3390/v14050856_

Round 1
Reviewer 1 Report
Dear authors,
congratulations on your manuscript. I have some suggestions to improve your work:
- References at square brackets must be before point at each sentence;
- Use verbs at past form;
- At line 43, terms "silent" and "occult" did were not well explained. Substitute for a single term;
- Be careful when you write "studies" and include only one reference. This situation happened between lines 43 and 46;
- At line 50, you must detail HBV DNA. You should write "partially double-stranded circular DNA";
- At line 65, substitute term " small numbers";
- At line 86, substitute term "HCV load" for "HCV viral load";
- At line 112, reagent's name is "QIAmp Viral RNA Mini kit"? Please modify term "Quiagen" for "Qiagen";
- You should review the explanation about molecular tests at topic 2.4. What kit did you use for one-step RT-PCR? What is reaction final volume? You include volume as mL, is it correct?
- For HCV genotyping, real-time PCR and/or conventional sequencing has been done? RFLP is not a gold-standard technique;
- At line 135, you should specify that PCR assays are real-time PCR, right?
- At lines 186, you should include absolute number in parenthesis after 9.87%. That percentage was not exposed in results;
- At lines 186-188, is it correct to associate occult HBV infection with anti-HBc?
- At line 189, I don't think it is correct to discuss your results with a previous study from patients undergoing hemodialysis;
- At lines 209-212, your study and study from Marot et al had corroborated. You should discuss a hipothesis for this finding.
Author Response
Response to Reviewer 1 Comments
Point 1. References at square brackets must be before point at each sentence;
Response 1: It was accepted and changed throughout the article.
Point 2. Use verbs at past form;
Response 2: It has been changed throughout the article.
Point 3. At line 43, terms "silent" and "occult" did were not well explained. Substitute for a single term;
Response 3: The text has been corrected. The terms “silent” or “occult” have been excluded. The work of Gonçalves et al 2019 indicates the existence of mistakes in the notification of Hepatitis B and C cases in the north region of Brazil. This compromises the prevalence rates of hepatitis B and C in the north region.Point 4. Be careful when you write "studies" and include only one reference. This situation happened between lines 43 and 46;
Response 4: It has been changed throughout the article.
Point 5. At line 50, you must detail HBV DNA. You should write "partially double-stranded circular DNA";
Response 5: It has been changed throughout the article. “Hepatitis B is caused by the hepatitis B virus (HBV), which belongs to the family Hepadnaviridae and whose genetic material consists of a with a double-chained circular DNA measuring 3.2Kb that replicates through a process that involves an RNA intermediate and reverse transcription [7].”
Point 6. At line 65, substitute term " small numbers";
Response 6: The term was replaced by low viral detection.
Point 7. At line 86, substitute term "HCV load" for "HCV viral load";
Response 7: The term has been replaced as requested.
Point 8. At line 112, reagent's name is "QIAmp Viral RNA Mini kit"? Please modify term "Quiagen" for "Qiagen";
Response 8: The term has been changed as requested.
Point 9. You should review the explanation about molecular tests at topic 2.4. What kit did you use for one-step RT-PCR? What is reaction final volume? You include volume as mL, is it correct?
Response 9: For the development of the one-step RT-PCR, the SuperScript™ III One-Step RT-PCR System with Platinum™ Taq DNA Polymerase kit (Invitrogen, USA) was used. The volumes used in the reactions were corrected in the text of the article.
Point 10. For HCV genotyping, real-time PCR and/or conventional sequencing has been done? RFLP is not a gold-standard technique;
Response 10: For HCV genotyping, the restriction fragment length polymorphism (RFLP) technique using AvaII and RsaI restriction enzymes was used. According to the methodology described by Hazari et al 2004 [19].
Point 11. At line 135, you should specify that PCR assays are real-time PCR, right?
Response 11: The real-time PCR technique was used. The text has been corrected by adding qPCR.
Point 12. At lines 186, you should include absolute number in parenthesis after 9.87%. That percentage was not exposed in results;
Response 12: This point was corrected and inserted in the results text. "the coinfected group consisted of the 32/324 (9.88%) patients".
Point 13. At lines 186-188, is it correct to associate occult HBV infection with anti-HBc?
Response 13: The association of occult hepatitis with the anti-HBc marker is incorrect. The text has been corrected.
Point 14. At line 189, I don't think it is correct to discuss your results with a previous study from patients undergoing hemodialysis;
Response 14: The text was wrong. The article by Sant’Anna et al demonstrates the prevalence of occult hepatitis B among patients with suspected HBV infection in the city of Belém, Pará. The text was corrected in the article.
Point 15. At lines 209-212, your study and study from Marot et al had corroborated. You should discuss a hipothesis for this finding.
Response 15: The text was corrected in the article.
Reviewer 2 Report
In my analysis, the study is relevant and interesting, methodologically well conducted.
It needs minor spelling adjustments and checks of the sum of the percentages to total 100% in the percentages of the results.
The discussion can be “clearly”, without repeating the p-values ​​already presented in the results, to dynamize the reading of the text.
I suggest that the first sentence of the conclusion be deleted or moved to another part of the text, to make the conclusion more punctual according to the work.
These adjustments do not invalidate the importance of the study.
Author Response
Response to Reviewer 2 Comments
Point 1. It needs minor spelling adjustments and checks of the sum of the percentages to total 100% in the percentages of the results;
Response 1: Spelling adjustments were made and the values presented in the tables were recalculated so that the sum reached 100%.
Point 2. The discussion can be "clear", without repeating the p-values already presented in the results, to streamline the reading of the text;
Response 2: The p-values described in the discussion were removed.
Point 3. I suggest that the first sentence of the conclusion be deleted or moved to another part of the text, to make the conclusion more punctual according to the work;
Response 3: This sentence has been removed from the text.

Reviewer 3 Report
In the manuscript “Comparison of Patients Monoinfected with Hepatitis C Virus and Coinfected with Hepatitis B/C in the Amazon Region of Brazil”, Sampaio et al. analyzed the HBV/HCV co-infection rates in a Brazilian population with high endemic, evaluating their characteristics and risk factors. The topic has been extensively addressed in the literature, with much larger numbers. Therefore, the topic seems non-original and does not provide substantial news, beyond epidemiological data useful for the specific local community.
Furthermore, some changes and/or clarifications appear necessary. Specifically:
- The Authors indicate that 13 HBsAg-positive patients are HBcAb-negative. The data is unusual. The Authors are asked to provide a hypothesis and compare the data with what is present in the literature.
- The discussion seems not exhaustive and underlines the poor content of the manuscript. It could be useful to broaden the discussion, in particular regarding the comparison with the literature of the data on the lack of association between hepatic fibrosis and coinfection.
- In the paragraph "Introduction", dubious estimates of the global prevalence of viral hepatitis are indicated. It is advisable to check these estimates. The following reference is useful: Polaris Observatory HCV Collaborators. Global prevalence and genotype distribution of hepatitis C virus infection in 2015: a modelling study. Lancet Gastroenterol Hepatol. 2017 Mar;2(3):161-176.
- There are apparently superfluous parts in the text. It is recommended to synthesize or eliminate:
- Lines 47-51
- Lines 96-105 (also indicate the relative reference)
- Lines 113-125
- In line 61, it is indicated that in co-infected patients there is a high frequency of positivity for HCV markers. However, all HCV-infected patients have related positive markers.
- In subparagraph 2.1 HIV infection is indicated as an exclusion criteria. In my opinion, this criteria can generate a selection bias and potentially alter the study results, in particular the assessment of risk factors for coinfection. Please explain your choice and correct if necessary.
- In table 2, the use of alcohol or tobacco is indicated as "never / rarely" or "frequently". It is advisable to specify in a note what is meant by these terms.
For all these reasons, I consider this manuscript eligible for publication after a few revisions. However, the final choice is referred to the Editor, in relation to the lack of originality and significant innovations brought by the manuscript to international literature.
Author Response
Response to Reviewer 3 Comments
Point 1. In the manuscript "comparison of patients monoinfected with hepatitis c virus and coinfected with hepatitis B/C in the Amazon region of Brazil", Sampaio et al. analyzed the rates of HBV/HCV coinfection in a Brazilian population with high endemicity, evaluating its characteristics and risk factors. The topic has been widely discussed in the literature, with much larger numbers. Therefore, the topic seems unoriginal and does not bring substantial news, other than epidemiological data useful for the specific local community;
Response 1: In Brazil, there are few studies that evaluate the coinfection between the HBV and HCV viruses. In the northern region of Brazil these studies are scarce. Brazil is a country of continental dimension with great diversity between different regions. The state of Pará is part of the Amazon region and has a socioeconomic, cultural and genetic reality that is different from other regions. Influenced by the indigenous ancentricity that is still present in our society. The work is original because it addresses the issue of coinfection in an Amazonian population.
Point 2. The authors indicate that 13 HBsAg positive patients are HBcAb negative. The data is unusual. Authors are asked to present a hypothesis and compare the data with what is present in the literature.
Response 2: The serological profile of reactive HBsAg and non-reactive anti-HBc can be found in patients at the beginning of the acute form. The immunological window for those against the viral core is approximately 45 days, after the outbreak. receipt of HBsAg. Additionally, DNA-HBV was isolated from serological samples from these 13 patients, which confirms HBV infection.
Reference: Brasil. Ministério da Saúde. Secretaria de Vigilância em Saúde. Departamento de Vigilância, Prevenção e Controle das Doenças Sexualmente Transmissíveis, Aids e Hepatites Virais. Manual Técnico para o Diagnóstico das Hepatites Virais. 2ed. Brasilia. Ministério da Saúde 2018.
Point 3. The discussion does not seem exhaustive and underlines the poor content of the manuscript. It may be useful to broaden the discussion, especially with regard to the comparison with the literature of data on the absence of an association between hepatic fibrosis and coinfection.
Response 3: The comment was accepted and the discussion on liver fibrosis and coinfection was expanded.
Point 4. In the "Introduction" paragraph, dubious estimates of the global prevalence of viral hepatitis are indicated. It is advisable to check these estimates. The following reference is useful: collaborators of the Polaris HCV observatory. Global prevalence and genotype distribution of hepatitis c virus infection in 2015: A modeling study. Lancet Gastroenterol Hepatol. 2017 Mar ;2 (3):161-176.
Response 4: Estimates have been updated according to the suggested benchmark.
Point 5. There are apparently superfluous parts of the text. It is recommended to synthesize or eliminate:
Lines 47-51
Lines 96-105 (also indicate relative reference)
Lines 113-125
Response 5: These parts were summarized in the text.
Point 6. In line 61, it is indicated that in coinfected patients there is a high frequency of positivity for HCV markers. However, all HCV-infected patients have related positive markers.
Response 6: This paragraph has been rewritten and additional information has been added for understanding.
"In HBV/HCV co-infection, viral serologies vary depending on whether the co-infection is simultaneous or superinfection, but HCV infection is almost always ev-identified by the detection in the serum of HCV RNA and antibodies against the infection ( anti-HCV) [13], while HBV infection can be evident or occult."
Point 7. In subparagraph 2.1, HIV infection is indicated as an exclusion criterion. In my opinion, this criterion can generate a selection bias and potentially change the results of the study, especially the assessment of risk factors for coinfection. Please explain your choice and correct if necessary.
Response 7: The aim of this study was to describe and compare the epidemiological and laboratory profile and the degree of hepatic fibrosis between HCV-monoinfected and HBV/HCV-coinfected patients in the Brazilian Amazon region. The presence of a third virus, HIV, which can compromise the host's immune response would lead to changes in laboratory tests, making it impossible to compare HBV and HCV.
Point 8. In table 2, the use of alcohol or tobacco is indicated as "never/rarely" or "often". It is advisable to specify in a note what is meant by these terms.
Response 8: For better understanding, we eliminated the aforementioned terms and added the terms "yes" and "no" to the table, and in the methodology the criteria adopted for this classification according to the literature.
The standardized questionnaire regarding lifestyle habits (excessive alcohol consumption and smoking habits) was used. Average drinking levels were evaluated to calculate the amount of alcohol consumed: sugarcane rum = 40%; beer = 5%; wine = 12%; Other Distilled Beverages = 55%. The threshold of alcohol consumption was standardized by calculating daily average intake in grams with an acceptable dose of 70 g alcohol per week for women and 140 g for men [16].
Smoking intensity was defined as the total number of cigarettes consumed per day, with each hand-rolled cigarette being equivalent to five manufactured cigarettes [17]. Subjects smoking more than five cigarettes per day were classified as smoke-ers and those smoking fewer than five cigarettes per day or did not smoke at all were classified as non-smokers.

Round 2
Reviewer 1 Report
Dear authors,
I feel satisfied with your responses.